# 3D Food Printing Applications Related to Dysphagia: A Narrative Review

**DOI:** 10.3390/foods11121789

**Published:** 2022-06-17

**Authors:** Tim Lorenz, Michèle M. Iskandar, Vahid Baeghbali, Michael O. Ngadi, Stan Kubow

**Affiliations:** 1School of Human Nutrition, McGill University, Montreal, QC H9X 3V9, Canada; tim.lorenz@mail.mcgill.ca (T.L.); michele.iskandar@mcgill.ca (M.M.I.); 2Department of Food Hygiene and Quality Control, School of Nutrition and Food Sciences, Shiraz University of Medical Sciences, Shiraz 71348-14336, Iran; baeghbali@shirazu.ac.ir; 3Department of Bioresource Engineering, McGill University, Montreal, QC H9X 3V9, Canada; michael.ngadi@mcgill.ca

**Keywords:** dysphagia, 3D printing, nutrition, texture modification

## Abstract

Dysphagia is a condition in which the swallowing mechanism is impaired. It is most often a result of a stroke. Dysphagia has serious consequences, including choking and aspiration pneumonia, which can both be fatal. The population that is most affected by it is the elderly. Texture-modified diets are part of the treatment plan for dysphagia. This bland, restrictive diet often contributes to malnutrition in patients with dysphagia. Both energy and protein intake are of concern, which is especially worrying, as it affects the elderly. Making texture-modified diets more appealing is one method to increase food intake. As a recent technology, 3D food printing has great potential to increase the appeal of textured foods. With extrusion-based printing, both protein and vegetable products have already been 3D printed that fit into the texture categories provided by the International Dysphagia Diet Standardization Initiative. Another exciting advancement is 4D food printing which could make foods even more appealing by incorporating color change and aroma release following a stimulus. The ultra-processed nature of 3D-printed foods is of nutritional concern since this affects the digestion of the food and negatively affects the gut microbiome. There are mitigating strategies to this issue, including the addition of hydrocolloids that increase stomach content viscosity and the addition of probiotics. Therefore, 3D food printing is an improved method for the production of texture-modified diets that should be further explored.

## 1. Introduction

Dysphagia is a condition in which some part of the swallowing mechanism is impaired [1]. It can cause coughing or choking due to abnormal delays in food bolus movement during swallowing, which disrupts the swallow initiation, as a part of food remains in the oral cavity. This can lead to reduced food consumption, weight loss and nutritional deficiencies [2]. Such complications in the diet are currently addressed mainly through the texture modification of foods and liquids [1,3]. The tolerable texture is unique to the patient but foods and liquids generally need to be soft and cohesive to ease bolus formation [3]. The bolus is a ball-like product of chewing and needs to be of a specific texture for the dysphagic patient. Three-dimensional food printing is a recent technology that holds great promise in improving this treatment method because it can enhance the visual appeal and deliver greater consistency of the produced food while reducing the caretaker’s load in making it [4].

Dysphagia can vary greatly in severity. It can be caused simply by the weakening of the muscles involved in swallowing because of aging [3]. More severe manifestations occur when dysphagia is caused by other diseases, especially stroke [1,3]. Other diseases that can also lead or contribute to dysphagia include Parkinson’s disease, Alzheimer’s disease, motor neuron disease, and gastroesophageal reflux disease (GERD) [1,5]. Finally, it can be caused by anatomic abnormalities or mechanical obstruction in the mouth [1].

When dysphagia is severe, texture-modified diets among other treatments are necessary to ensure the safety of the patient [3]. Texture-modified diets refer to diets where the food is manipulated into a texture that the patient is more likely to swallow safely [3]. If they are not properly administered, the patient can suffer from aspiration, which is the movement of the food from the mouth into the airway. This can cause coughing, choking, or aspiration pneumonia [1]. Aspiration pneumonia is a result of silent aspiration where the food or liquid moves into the airway without causing a coughing response. This can be fatal if it goes unnoticed [1].

While texture-modified diets are necessary, they bring with them several complications for the patient. In general, foods that are served soft after boiling or mashing are visually unappealing and have their taste and sometimes nutritional value diluted [4,6,7]. This nutritional dilution occurs when water is added to foods, or they are excessively boiled for texture adjustment [4]. The addition of water results in a higher volume without increasing the nutrient content, so there is a reduction in energy and nutrient density [4]. Boiling foods in water decreases the mineral and carbohydrate content in foods through leaching, while protein content is maintained [8,9,10]. This means patients could seemingly eat an adequate amount while lacking some micronutrients.

An additional limitation of texture-modified diets is that the mealtime environment needs to be optimized for patients to ensure concentration on the eating process. This includes no conversation during eating and cleaning of the mouth prior to every meal to reduce bacteria that can cause pneumonia. These factors make eating a chore and diminish its social quality, which reduces appetite in many patients [1]. Overall, this leads to patients on texture-modified diets having a lower intake of both energy and protein, resulting in malnutrition (see Table 1) [11].

A pilot study by Farrer et al. [12] already showed that the lack of appetite can be improved by enhancing the visual appeal of the food. This was achieved by using food molds, which take in the pureed food material and produce a 3D shape, usually a representation of what the food looked like before pureeing. Farrer et al. [12] showed that the proportion of patients who finish 75% to 100% of their meal compared to 0–25% of their meal is higher in the group presented with molded foods. Another measurement, the median plate waste, was not statistically different in the small sample of the pilot study, but a Cohen’s D size effect test indicated that the results should be studied in a larger cohort where the results may prove significant [12]. Germain et al. [13] confirmed this tendency in a similar study, showing a significant increase in food intake. The treatment group eating molded foods managed to increase weight despite experiencing involuntary weight loss before. They improved intakes of energy, protein, various minerals, and vitamin D as well as vitamin B_2_. Their study was performed on a small sample of 17 participants, so although interesting, the results were not generalizable. This is also because the population was heterogeneous and many participants were not able to give any feedback, due to age-related mental decline [13]. An additional reason why molded foods increased food intake was that caretakers gave enthusiastic feedback about it, which may have caused them to encourage the patients to eat more food [13].

In this regard, the 3D printing of food holds even greater potential than food molding. Three-dimensional printing is an additive manufacturing technology, in which usually successive layers of material are deposited onto the previous layer [6]. This can be done with various techniques, with extrusion-based printing being the current relevant one for dysphagia food. Here, the food material or ink is deposited using an extruder on an initial surface and then successively layered [6]. For this to be successful, the material needs to be soft during extrusion and solidify once deposited to support the next layer [6]. Edible materials that possess this property are hydrocolloids, which are long polymeric chains that can form 3D networks in aqueous solutions through van der Waals forces, hydrostatic interactions, and/or hydrogen bonding. This makes the solution they are suspended in viscous [14]. To be beneficial for extrusion-based printing, the hydrocolloid must confer shear-thinning properties to the food [6,15,16]. This means that the hydrocolloids lose their interactions when they experience shear stress at the printer nozzle, making them fluid. They then regain their interactions when that stress is released after deposition, causing them to make the solution viscous once again (see Figure 1) [16].

The reason that 3D-printed foods have a high potential to improve texture-modified diets is that molded foods often have a gelatinous mouthfeel because starch is mainly added to retain the shape of the food, starch being a hydrocolloid [15]. This is not as necessary for 3D-printed foods due to the more extensive study of different hydrocolloids and their mixtures [15]. To render a food 3D printable, three research steps are generally needed when it comes to hydrocolloids [16]. First, several hydrocolloids are considered. Second, their optimal ratio is found. Third, the minimal amount of hydrocolloid needed is determined [16]. This ensures the printability and dysphagia suitability of the product with the least amount of flavor or texture alteration. This extensive research is not done for molded foods, and often starch is used which can result in an overly gelatinous mouthfeel [15]. Dysphagia-friendly foods that have already been created using this method are 3D-printed pork, beef, peas, carrots, and bok choy [15,17,18].

Three-dimensionally printed foods also require less handling than molded foods, which can improve safety and lessens the burden on staff [15]. Additionally, 3D food printers can produce internal structures because the final product is constructed layer by layer. This means the 3D printer draws successive 2D layers, which allows for the outside surface to be a cohesive whole, while the internal structure can be constructed variably. This significantly contributes to the stability, texture, and density of the food [15]. Finally, with the right printer that allows for new designs, the possibilities of shapes are virtually endless, whereas molds are limited by storage space [15]. In fact, Kouzani et al. [7] produced a 3D-printed meal of pureed pumpkin, beetroot, and tuna. This was judged to have the same flavor as a meal of pureed pumpkin, beetroot, and tuna prepared by a chef who made it for the study. The chef and two other people compared the two meals and judged them to be the same in flavor [7]. The same cannot be said for molded foods, due to the aforementioned addition of starch, causing excess gel formation in the food [15]. Something else to consider may be the higher expectations for molded foods, due to their higher visual appeal, which are then unmet, as they taste the same or worse than their pureed counterparts [12]. This effect may also apply to 3D-printed foods but has yet to be studied with the help of patient feedback.

Besides the advantages of less handling, internal structures, and better or similar taste that 3D-printed foods have over molded foods, 3D food printing has some more benefits with respect to texture-modified diets. One potential benefit is that of personalized nutrition; this is the addition of certain nutrients or non-traditional ingredients high in certain nutrients [4]. This can be done conveniently with 3D printing since the food is homogenous prior to printing and any special ingredients can be added and mixed throughout [4]. For texture-modified diets, the need for skilled chefs and kitchen workers would be reduced in any care facility since the food would only have to be printed on-site or microwaved if delivered [7]. One of the most significant benefits of 3D food printing is the consistency of the product. Three-dimensional food printers can almost always provide the correct texture for the patient, and this will reduce fatal choking and aspiration pneumonia [4].

Overall, 3D food printing has a huge potential to influence the food industry. In its current state of technology, it will mostly benefit those living on a texture-modified diet [4]. There, it has the potential to improve nutrient intake by making food that is more appealing and a larger variety of it (Figure 2). This food is also more easily produced, reducing the burden on staff and making it accessible to more patients. It also has the potential to improve the nutritional value of the diet through convenient fortification before processing [4]. Because the potential for human error in producing incorrectly textured meals is reduced, the printed food is safer [4]. Drawbacks that need to be considered are the loss of nutritional value through additional processing and the patient’s disappointment if the food flavor does not live up to its visual appeal.

More research must still be done to properly implement an exclusive or partial 3D printed diet for patients with dysphagia. A greater variety of dysphagia-friendly foods needs to be produced, and food safety protocols need to be created [15]. How the additional processing, such as heat treatment and extrusion, affects the nutritional value of these products is another avenue for research [19]. Finally, and most importantly, how 3D food printing affects the nutritional status and clinical outcomes of people with dysphagia needs to be studied since the ultimate goal is to improve their food intake and quality of life [19]. In the following review, the current state of 3D food printing as it pertains to texture-modified diets will be discussed.

## 2. Three-dimensional Food Printing

Three-dimensional printing is a layer-by-layer manufacturing process that can create semi-solid or solid geometrically complex shapes. It has the potential to feasibly customize the sensorial and nutritional properties of 3D-printed foods. The main challenges of 3D printing foods include, but are not limited to, the printability of ingredients and macronutrients, printing parameters, slicing and model creation, and printing equipment [20]. When it comes to 3D food printing, there are four different possible techniques [6]. Two of the techniques rely on an initial bed of a powdered substance. In selective sintering printing, a heat source, such as a laser or hot air, is applied precisely to the powder bed. This causes the powder to melt, bind, or fuse with the surrounding material and then harden after it cools [6]. Another layer of powder is added, and the process is repeated until the 3D object is created [21]. Binder jetting printing is the other technique relying on a powder bed. In this technique, tiny drops of a binder are added to the powder, which liquifies it, allowing it to fuse and form a temporary structure. Subsequently, heat is applied to dry the structure, which stops the reaction and solidifies it [6]. Again, a new layer of powder is applied, and the process is repeated [21]. Both methods have disadvantages that have so far made them unsuitable for research on dysphagia foods. They can only produce solid foods because leaving them in a liquid state would cause reactions with the surrounding powder, so no 3D structure could be created [6]. Additionally, these two techniques can only use powders, such as starch, sugar, or proteins as their material, so a complete nutritious meal cannot yet be created from them [6,21].

Another technique is inkjet printing, in which a liquid is deposited on an already existing food product [6]. While having high precision, it is only able to make 2D graphical decorations or fill in cavities of solid foods with a liquid [6,21]. The current reliance on a solid foundation and having no options for texture alteration make this method inapplicable to dysphagia foods. In summary, these three techniques are not currently viable to produce texture-modified foods for patients with dysphagia.

Moreover, extrusion-based printing, the fourth 3D food-printing technique, is very suitable for dysphagia foods, giving it all of the research attention in this field, and in fact most of the attention in 3D food printing [19]. The technology for it was adapted from fused deposition modelling, where plastics are heated above their melting point, deposited on a surface, and built up layer by layer [6,21]. For food printing, a movable extruder nozzle deposits the food material in the form of viscous liquids that are thin enough to be extruded and thick enough to retain their shape once deposited [6,21]. The printer is loaded using cartridges containing the viscous food mixture, which is why they are also referred to as food ink [18]. The ink is extruded out of the nozzle with the force of a piston and deposited on the printing surface as designed in a printing software [21]. This printing technology requires the use of foods that are viscous but not too thin, which makes it the perfect candidate for the 3D printing of dysphagia-friendly foods.

### 3D Food Printing for Dysphagia

When it comes to producing dysphagia foods, it is useful to categorize them based on texture because patients with dysphagia have varying texture tolerances [3]. Textures that are generally used in acute or long-term care settings are described by the International Dysphagia Diet Standardization Initiative (IDDSI) (see Figure 3) [22]. This framework is also referred to in the 3D food printing for dysphagia literature [15,17,18]. The IDDSI framework organizes foods and liquids on a continuous, descending order of 8 (7–0) textures. Foods can be on levels 7–3, and liquids can be on levels 4–0, so there is an overlap between the thinnest foods and the thickest liquids [22]. A clinical swallowing ability evaluation is undertaken to see what texture level a patient can tolerate and what corresponding foods they will be able to swallow [3].

IDDSI also provides simple tests using household items so anyone can classify foods into the appropriate category [22]. These include spoon tilt, spoon pressure, and fork pressure tests [22]. The purpose is to test some of the important aspects of texture when it comes to dysphagia, which are hardness, cohesiveness, and adhesiveness [23]. The spoon tilt test examines whether the food slides off the spoon when tilted [22]. The food should stick together in this test, which is its cohesiveness [22]. The cohesiveness needs to be appropriate, meaning neither too high nor too low [23]. Foods that are too high in cohesiveness were found to accumulate in the pharynx [24], while foods with a low cohesiveness were found to scatter in the pharynx [25], both of which pose a risk for aspiration [23]. For a positive cohesiveness result, the food should hold its shape on the spoon and remain as one unit when sliding off [22]. Another factor that is tested is the adhesiveness of the food [22]. The adhesiveness should be low to avoid sticking to the pharynx, another aspiration risk [23]. An appropriate food falls off and either minimally or does not stick to the spoon when tilted by 90 degrees, with the option of flicking the food off the spoon one time [22]. The spoon and fork pressure test evaluates the hardness of the food and shows how easily the food breaks apart under the pressure of a spoon or fork [22]. How much pressure is needed is measured by whether or not blanching of the fingernail is observed [22]. The pressure that causes the fingernail to blanch is the same pressure that the tongue exerts during swallowing [15]. If the food breaks without nail blanching, the food can be categorized into the lower IDDSI levels of 4–5 [22].

Three-dimensionally printed foods that have been specifically produced for dysphagia patients are not yet very common. Kouzani et al. [7] were able to print a depiction of a fish using three pureed ingredients: tuna, pumpkin, and beetroot. They were mixed with water, cooked in a microwave oven, and then pureed. After this step, the printer barrels were loaded with the ink and extruded into the previously made, computer-aided design. Generally, pureed foods are one of the categories suitable for a dysphagia diet but a textural assessment using IDDSI methods needs to be performed to ensure the food’s suitability [7,22]. Considering this was the earliest work of 3D dysphagia food printing, the focus was more on the creation of the design in a software and the printing process using a Bioplotter extrusion printer [7]. This is a great study to build on, as it shows the possibility of printing pureed foods using extrusion. The use of creative software allows for endless food designs, the printer settings can be determined empirically, and the food ingredients are also exchangeable [7]. The almost limitless possibilities of 3D food printing when it comes to soft foods are one of its biggest strengths, and this study provides a blueprint of how to create new products [7]. The next step is to research texture alteration and perform texture assessment to ensure there are a variety of products suitable for any texture level of the IDDSI scale.

Pant et al. [18] studied the 3D printing of pureed fresh vegetables to produce dysphagia-suitable foods. They also made use of hydrocolloids at different concentrations and mixtures to produce optimal food inks. These should be printable, suitable for a dysphagia diet, contain a minimal amount of hydrocolloids, and not suffer from syneresis [18]. Syneresis is the reverse process of gelation, where the food product leaks water and loses structure during storage [18]. The vegetables studied were garden peas, carrots, and bok choy to represent ingredients on a spectrum of low to high water and starch content [18]. Similar to Kouzani et al. [7], the foods were cooked and pureed, but in the next step, had hydrocolloids at different concentrations added to them [18]. The hydrocolloids tested were xanthan gum, kappa carrageenan, locust bean gum, and their mixtures, all at relatively low concentrations of less than 2% *w/w* [18]. After printing all of the resulting inks, the optimal ink was determined based on print precision and syneresis measurements, with low separation being optimal [18]. When more than one ink per food was found to perform well, the lowest concentration of hydrocolloid was the deciding factor for the ink to go on to textural assessment [18]. The reason for choosing the lowest concentration of food hydrocolloid is that they can impart a non-natural flavor to the food, leading to lower consumer acceptance [18]. The optimal inks were able to pass IDDSI tests, meaning they were suitable for texture levels 7–5 [18].

Again, this research was able to provide a method of producing dysphagia-friendly foods from fresh vegetables that can be 3D printed into an aesthetically pleasing dish [18]. The use of hydrocolloids made it possible to print ingredients with high water content and to optimize the inks for printability, texture, and flavor [18]. It also provided a new method for 3D printing vegetables which were previously freeze-dried prior to printing. This allowed the authors to use less hydrocolloid and maintain more of the nutritional value and flavor of the vegetables [18].

Dick et al. [15,17] researched the printing of dysphagia-friendly meat. So far, they were able to print pork and beef, with the help of hydrocolloids. To give the meat a printable consistency, it was ground, mixed with water, and then mixed with the hydrocolloids. For the pork, any formulation that contained a mixture of the hydrocolloids tested, xanthan and guar gum, was suitable for printing and was compatible with the IDDSI levels 7–6 [15]. For beef, the hydrocolloids were necessary for 3D printing in the first place [17]. The hydrocolloids tested were xanthan gum, kappa carrageenan, locust bean gum, and guar gum. They were tested alone or in mixtures of two at a concentration of 0.5% or 1%. Given this many samples, suitable products were found for IDDSI levels 7–5 [17].

As seen above, hydrocolloids are a vital addition to making 3D food printing of some foods possible [6,16]. Here, it serves to bind any free water which prevents phase separation under shear during extrusion [17]. Besides this, they can also be added in varying amounts to alter the texture of foods, allowing for the different IDDSI levels of texture to be produced [15,17,18]. The hydrocolloids that are used for studying this are xanthan gum, guar gum, kappa carrageenan, and locust bean gum [15,17,18]. Xanthan gum is interesting because it does not form gels on its own and needs to be used in combination with a synergistic hydrocolloid, such as locust bean gum or guar gum, to form a gel [15,18]. Xanthan gum also repels hydrocolloids like starch, weakening the gel, which means that it can be used to decrease the viscosity of vegetable food inks [18]. For meat, no such repelling forces apply; xanthan gum interacts with soluble meat proteins to make a gel [15]. In general, hydrocolloids reduce the density of meat by retaining water that creates cavities in the tissue [15]. This reduces the hardness of meat, making it easier to chew, which is necessary for patients with dysphagia [15]. Any studied mixture of xanthan and guar gum was able to reduce the hardness of pork enough to make it compliant with IDDSI standards for dysphagia foods [15].

Whereas guar gum and xanthan gum are cold-swelling hydrocolloids, kappa carrageenan and locust bean gum are heat-soluble hydrocolloids, meaning that they only form a gel network after being heated [17]. This can be an issue while printing when the hydrocolloids are not yet activated. This means that they cannot bind the free water of the food ink causing phase separation during printing. This can result in a clogged printer nozzle when the solid particles of the ink accumulate [17]. On the other hand, heat-soluble hydrocolloids tend to form stronger gels leading to higher hardness of the printed food, which can be desirable depending on the starting ingredient and the patient the food is designed for [17].

In summary, a good foundation was set for the creation of 3D-printed dysphagia foods. Researchers such as Dick et al., Kouzani et al. and Pant et al. [7,15,17,18] were able to print foods that comply with IDDSI standards while also creating a methodology of empirically researching new food inks, making use of hydrocolloids where necessary.

Other advances that were made for the 3D printing of dysphagia foods is the creation of specific printer technology [26]. In the printer attachment designed by [26], a solid, commercial thickener is fed through an extrusion screw. At the same time, the liquid of choice flows into the extruder, where the two mix as they go through the extrusion screw. Then, the thickened fluid is extruded at the nozzle and deposited layer by layer. This results in a visually appealing, thickened liquid that can include several colors when the fluid is changed throughout the printing process, which is otherwise continuous. For example, a printed egg model can contain both white and yellow colors (see Figure 4). The researchers were able to achieve the predetermined viscosities of their liquids; higher viscosities also allow for more intricate designs because the structure is more solid [26].

The final 3D printing technology that could lead to more visually appealing dysphagia foods is 4D food printing, which has not been studied on dysphagia foods yet. This visual appeal could then increase appetite and lead to a greater food intake [12]. Four-dimensional food printing is defined as the physical or chemical change in an ingredient of the food product over time [27]. This ingredient, called a smart ingredient, reacts to environmental stimuli, human intervention, or internal stimuli [27]. This means that the food product needs at least one smart ingredient and the stimulus ingredient, which can either be added as a part of the food product, added externally after printing, or be part of the natural environment [27]. In the study of Ghazal et al. [27], the smart ingredient was an anthocyanin purple-red powder that was extracted from blueberries. The stimulus ingredients were chemicals that change the pH of water, such as citric acid, trisodium citrate, sodium bicarbonate, and sodium hydroxide, making different pH solutions ranging from 2 to 10 that were sprayed on the 3D printed anthocyanin gel. The other option was lemon juice, which was gelled and incorporated into the 3D-printed food as an internal stimulus. The researchers were able to achieve a color change when treating the product with low pH solutions. Spraying the product with the pH 2 solution, however, gave it an unnatural sourness. This led the lemon juice-incorporated sample to perform the best overall in a sensory evaluation following the ISO-4121 method [27]. Starch was found to lessen the color change, as it increased in concentration, so hydrocolloids must also be carefully considered for 4D food printing applications [27]. While this study was not designed to produce a dysphagia-oriented food, all products were essentially thickened liquids, making their application in a 3D-printed dysphagia diet interesting.

Flavor and aroma are other characteristics that can be influenced by 4D food printing to increase the product’s appeal [28]. This was studied by Phuhongsung et al. [29] who found new volatile compounds in their vanilla-flavored soy protein isolate gel. This change was observed after microwave heating and was attributed to the vanilla flavor releasing 1-octen-3-ol and ethyl maltol after a thermal reaction [29]. Additionally, the flavors of bitterness, astringency, umami, richness, and saltiness were more intense after microwave heating [29]. Using a pH spray to alter the flavor and aroma of 3D-printed foods is also an option, but this will only affect the surface of the product, whereas microwaving can affect the food throughout [28]. Considering that this product was a gel system, it could also undergo texture alteration and assessment to be potentially served to patients with dysphagia. The summary of advantages and disadvantages of 3D food printing for dysphagia are presented in Table 2.

## 3. Food Safety

During any food processing operation, the safety of the product needs to be ensured. This includes limiting microbial growth to acceptable levels and avoiding any food contact surfaces from leaching toxic chemicals [6]. Currently, the number of studies on food safety is very low, but Severini et al. [30] found a concentration of 4.28 log CFU/g in their printed smoothie, which warrants the sanitization of every food-contact surface before printing. Several heating and cooling steps in the production of many 3D printed foods, including before, during, and after printing, can be the cause of microbial growth [6]. Other potential issues that can arise are physical hazards that break off the printer, such as plastics or metals and food fraught due to the lack of resemblance of the product to the ingredients [19].

In general, 3D food printers must fulfill the same safety requirements that current food contact surfaces have [6]. This is the case for printers designed for food printing, but extra caution must be paid to generic 3D printers with food printing modifications [6]. While more research is necessary, the issue of food safety is not expected to be a major limitation for making 3D food printing a viable option [4].

## 4. Nutritional Aspects of 3D Food Printing for Dysphagia

There are some nutritional concerns when it comes to 3D food printing. The first is the highly processed nature of the resulting food product [4]. Getting the food into a printable state alone takes softening the food through boiling or steaming and then blending [18]. Boiling can cause the leaching of minerals and sugars [10], so steaming or microwave cooking would be preferred. Steaming does not have this leaching effect, as the food is not in direct contact with water [31]. In microwave cooking, the only water that is added is also used for the puree, so any leached nutrients are reincorporated [7]. Microwaving does, however, lead to the destruction of some nutrients, including vitamin C, chlorophyll, and soluble sugars and proteins [31], making steaming the best method for cooking vegetables before pureeing. Blending has to be done to ensure that the printing nozzle does not clog [18]. This means that the food must be able to pass through a certain sieve diameter depending on the nozzle size [18].

Many 3D-printed foods can be considered ultra-processed, given the lack of whole foods and the amount of food additives necessary, for example, thickeners [32,33]. A diet high in ultra-processed foods, which a 3D printed dysphagia diet would be, causes issues for the gut microbiome [32]. A lack of whole foods and a modulation in their dietary fiber and fats results in reduced microbial diversity, increased production of harmful compounds by the microorganisms, and modulation of the immune system, as well as consumption of host protein and mucosal compounds by the microorganisms. This results in inflammation [32]. Animal models show this effect to a lesser extent, even with high fiber, processed diets, meaning that processing foods result in altered metabolism of the nutrients, even if their composition remains the same [32]. One method of counteracting this effect is the consumption of probiotics, which are microorganisms that confer a health benefit to the host if consumed in adequate amounts [34]. Fortunately, probiotics can already be incorporated into 3D-printed foods, where they are maintained throughout processing in an amount that is adequate for them to exert their health benefits [34]. Liu et al. [34] produced 3D printed mashed potatoes containing *Bifidobacterium animalis* subsp. *lactis* BB-12, which has been shown to improve gastrointestinal health and immune function. The added amount of probiotics was only reduced when a small nozzle diameter of 0.6 mm or temperatures of 55 °C in the nozzle was used [34]. Such conditions were not reached in currently researched 3D-printed dysphagia foods. These use nozzle diameters from 0.84 mm to 1.50 mm and are printed at room temperature [15,17,18]. This means probiotics are a viable functional ingredient in the production of 3D-printed dysphagia foods. Moreover, the food researched by Liu et al. [34] was mashed potatoes, which could undergo texture alteration studies to make them suitable for patients with dysphagia.

Digestion is another factor that is highly influenced in various ways through processing and the addition of various ingredients [35]. This is highly relevant when developing foods for patients with dysphagia since the goal is to improve nutritional status, which is affected by the bioavailability of nutrients which in turn is affected by processing [35]. High heat treatment is generally associated with loss of nutritional value due to the Maillard reaction, where proteins interact with polysaccharides, making the affected amino acids unavailable [36]. Fortunately, foods created for patients with dysphagia cannot undergo high heat treatments and are generally processed in the presence of water [7,15,17,18]. This is because such processing conditions are opposite to the goal of making the food soft and thus easier to chew. On the other hand, cooking with water can cause protein denaturation or unfolding, which can increase their susceptibility to proteolytic enzymes in the stomach and small intestine [35,36].

As described above, gelling is used to alter the texture of dysphagia foods. For vegetables, this effect results in a higher puree viscosity and binding of free water [18]. Some hydrocolloids, including fiber and protein-based hydrocolloids, which are responsible for increasing viscosity and binding of water, will also increase the food’s viscosity in the stomach [35]. The higher viscosity of the stomach contents will lead to slower gastric emptying, making digestion occur over a longer period of time [35]. Some hydrocolloids require certain conditions to make them functional in increasing viscosity [17,35]. Therefore, they could be added to foods without activating them, meaning they would not contribute to the texture while eating but would thicken the stomach contents under acidic conditions. This leads to the previously described slowed gastric emptying and prolonged digestion [35]. This method improves the poorer nutrient absorption of processed foods relative to whole foods [35]. It could also be useful for dysphagia foods that are thickened with hydrocolloids, such as starch, which does not have the effect of increasing viscosity in the stomach, as it is easily hydrolyzed by amylase in the saliva and acidity of the stomach [35].

Three-dimensional printing itself involves pressure and sometimes heat which occurs at the extrusion nozzle [15,16]. The extrusion conditions in 3D printing could be described as being mild due to the high moisture content of the foods, low extrusion temperatures, and short times spent in the extruder [36]. This is because, for 3D food printing, extrusion is a method to deposit food ink at the correct position rather than processing food ingredients. In fact, 3D printing temperatures for dysphagia foods do not reach what is considered low in extrusion-based food processing [36]. The 3D printing of dysphagia foods generally happens closer to room temperature [15,17]. This means that nutrients, including all vitamins and minerals, are likely retained during 3D printing. Non-nutrient health components, such as phenolic compounds, including flavonoids, isoflavones, and anthocyanins, tend to decrease under all extrusion conditions, but this was tested on the processing of foods via extrusion rather than 3D printing [36]. This means their retention would be an interesting avenue of research for extrusion printing.

One method to counteract these nutrition concerns is the active removal of processing steps in the production of 3D printed dysphagia foods. In the study by [18], one major aspect was achieving a 3D-printed product using fresh vegetables. Prior to this, the 3D printing of vegetables involved freeze-dried materials with added water and hydrocolloids [18]. Freeze-drying causes the destruction of vitamin C, vitamin E, and folic acid [18]. Another method for decreasing the nutritional losses from processing is the fortification of foods prior to printing [4]. This is also the concept of personalized nutrition, where nutrients are added to foods, or ingredients with special nutritional constituents are used in place of traditional ingredients [4]. One easy example of this would be to use milk instead of water when preparing a puree for extrusion [4]. This would have the same functionality of making the puree thinner, but instead of diluting the nutritional content, the milk would add energy, protein, and micronutrients to the food [4].

While functional food development using personalized nutrition is still lacking for patients with dysphagia, many products have already been developed for the general public that can inspire new dysphagia food research. Some of the food development focuses on creating a dough that is then baked, making it less applicable for dysphagia foods. This was the case for Krishnaraj et al., (2019) [37], who extruded a dough made of a composite flour using indigenous legumes, grains, and seeds chosen for their high fiber and protein contents. Hydrocolloids were added to the dough to improve printability. Post-processing using deep frying, hot air drying, and microwave drying resulted in various snacks, each with different qualities [37]. While this idea is unlikely to be useful for patients with dysphagia, Wilson et al. [38] used the same composite flour to create 3D printed, fiber-enriched chicken. This could be a great dysphagia-friendly product, since the 3D printing of meat was already proven to be a possibility in creating dysphagia foods by using the properties of hydrocolloids [15,17]. Another dough was enriched with dried larvae and yellow mealworms to increase protein content [39]. Such functional and sustainable ingredients could also be added to dysphagia-friendly foods. Varghese et al. [40] created a dough using jackfruit seed powder, finger millet powder, butter, soy protein isolate, and a vitamin and mineral premix as the main constituents. The researchers were able to influence the hardness of the cookies after baking by adjusting temperature, water to butter ratio, and how much dough was used on the inside of the cookie, meaning that texture alteration to dysphagia-suitable levels may be possible given more research [40].

Many other 3D-printed functional foods are soft, extruded gel systems similar to dysphagia foods, so one could easily imagine that they could be made into foods that comply with IDDSI regulations with only a small amount of research. Functional ingredients that have already been printed include cordyceps flower powder [41]; various protein powders, hawthorn, spinach, purple sweet potato, and lemon powder [42]; mushrooms, white beans, dried non-fat milk, and lemon juice [43]; and pure orange fruit concentrate [44]. Making various products with these ingredients and other not yet researched healthful ingredients could make a healthy, varied 3D-printed diet for patients with dysphagia possible. Additionally, optimizing them for digestion by using the right hydrocolloids and processing procedures could ensure that all the desired nutrients are accessible to the consumer. A summary of nutritional concerns and suggested alleviation methods for 3D-printed dysphagia foods are presented in Table 3.

## 5. Future Research

A major necessity for 3D food printing for dysphagia research is to develop a variety of foods for every IDDSI texture level. This has not yet been achieved for the printing of meat, meaning that patients on texture level 4—pureed—would be missing a protein option in their meal [15,17,22]. User-centered research, including sensory evaluation by patients, should also be incorporated to ensure consumer acceptability [19]. Once achieving a variety of foods for every texture level, ensuring food safety and establishing food safety protocols are vital before being able to serve the 3D-printed foods to patients [15]. Once safe food options are provided, another avenue of research is the nutritional content of 3D printed foods before and after printing, as well as the most important metric of whether the nutritional and health status of patients with dysphagia is improved by providing 3D-printed visually appealing and consistently safe foods [19].

## 6. Conclusions

This review was concerned with the 3D printing of foods suitable for dysphagia patients. Dysphagia is a condition in which the swallowing mechanism is impaired. As a treatment, the texture of foods and liquids is altered to make them easy to swallow. Which texture a patient is prescribed depends on the severity of their condition. With the recent advances in 3D food printing, texture-modified foods specifically designed for patients with dysphagia are starting to be explored.

The most promising feature of 3D food printing is the visual appeal that 3D-printed foods can have without the need for skilled chefs. This increased visual appeal will hopefully increase the appetite of patients on texture-modified diets, who often suffer from malnutrition due to the unappealing nature of mashed and pureed foods. Currently, four 3D food printing technologies exist, including selective sintering printing, binder jetting printing, inkjet printing, and extrusion printing, with only the latter being useful for dysphagia foods as of now.

With this technology, two types of meat (pork and beef) and three vegetables (peas, carrots, and bok choy) have been printed. These foods were also assessed for texture and were appropriate for patients with dysphagia. This was achieved with the appropriate addition of hydrocolloids, which are water-binding compounds and gelling agents. The authors of these studies also outlined their method for producing these foods, so more 3D printed, dysphagia-suitable foods can be produced more easily. Whether these foods increase patient food intake and improve their nutritional status remains to be researched. Other advances that have been made are the creation of 3D printer attachments for thickened liquids, and 4D printing, in which the food product becomes more appealing over time after being exposed to a stimulant.

How this extent of processing, including freeze-drying, cooking, mashing, pureeing, sieving, and extruding, affects the nutrient content, digestibility, and health of the gut microbiome of the consumer is a concern. Work that has been done to improve this issue is the removal of processing steps, the addition of probiotics, and personal nutrition.

## Figures and Tables

**Figure 1 foods-11-01789-f001:**
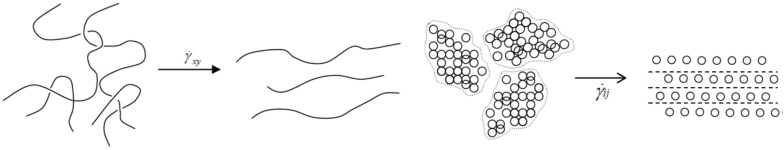
Shear-thinning mechanism in food systems. Disentanglement of long macromolecular chains (**Left**). Breakdown of loose particle clusters to form layered alignment (**Right**) (Reprinted from Tan et al., 2018 [16]. Permitted by Creative Commons Attribution-Non Commercial 4.0 International License http://creativecommons.org/licenses/by-nc/4.0/, accessed on 6 June 2022) [16].

**Figure 2 foods-11-01789-f002:**
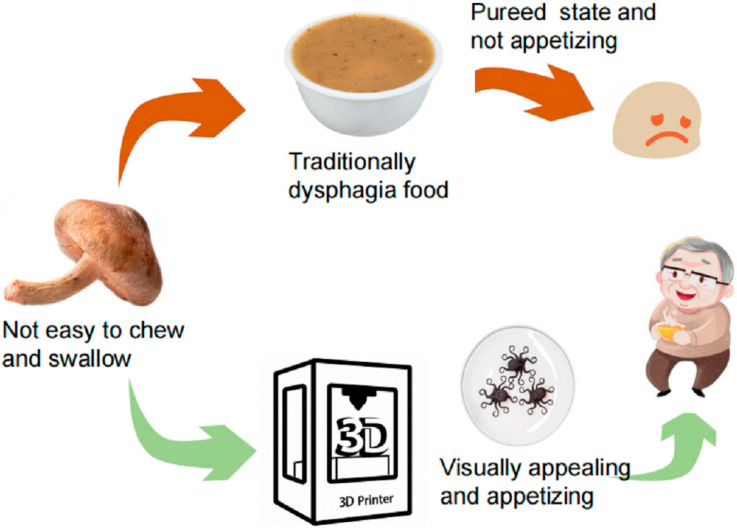
How 3D printing can help create more appealing dysphagia foods (Reprinted from Liu et al., 2021 [2]. Permitted by Creative Commons Attribution-Non Commercial 4.0 International License https://creativecommons.org/licenses/by/4.0/, accessed on 6 June 2022) [2].

**Figure 3 foods-11-01789-f003:**
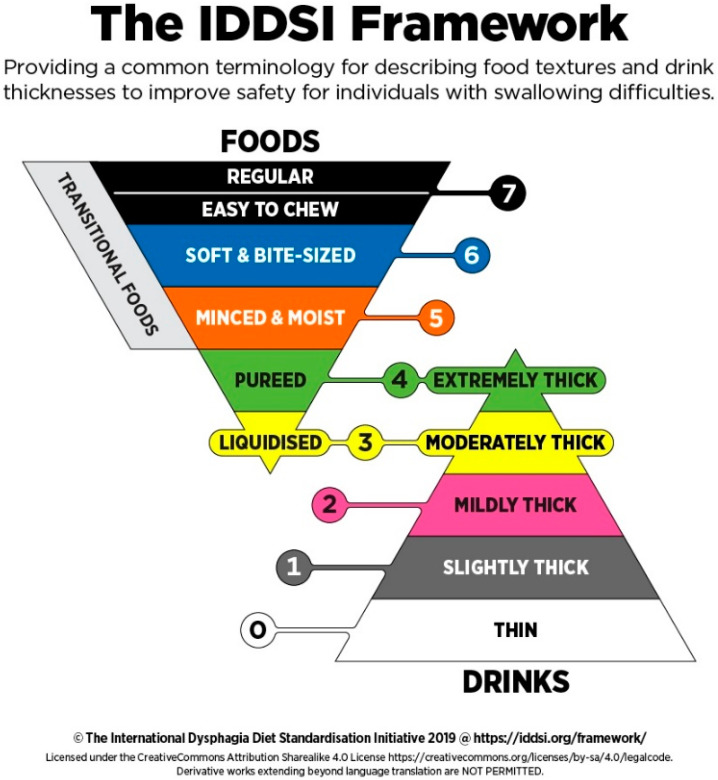
The IDDSI framework (Reprinted from IDDSI, 2019 [22]. Permitted by Creative Commons Attribution-Non Commercial 4.0 International License https://creativecommons.org/licenses/by/4.0/, accessed on 6 June 2022) [22].

**Figure 4 foods-11-01789-f004:**
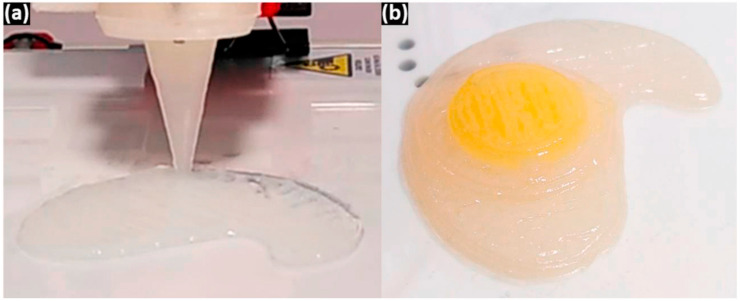
Printing of a simulated fried egg using thickened milk and orange juice showing (**a**) 3D printing nozzle (**b**) the 3D printed product (Reprinted with permission from Diañez et al., 2021 [26]. © 2021, Elsevier Ltd., Amsterdam, The Netherlands) [26].

**Table 1 foods-11-01789-t001:** Energy and protein requirements and intake [11].

	Normal Diet Group (SD)	Dysphagia Diet Group (SD)	*p*-Value
Energy requirements (kJ)	6472 (882)	6426 (1029)	0.79
Protein requirements (g)	66 (14.2)	62 (13.1)	0.29
Energy consumed (kJ)	6115 (2575)	3877 (1420)	<0.0001
Protein consumed (g)	60 (27)	40 (18.6)	0.003
Energy deficit (kJ)	357 (2366)	2549 (1066)	<0.0001
Protein deficit (g)	6 (24.8)	22 (16.9)	0.013

**Table 2 foods-11-01789-t002:** Summary table of advantages and disadvantages of 3D food printing for dysphagia.

Advantages	Disadvantages
Foods of many classes can be successfully printed, including protein, vegetable, starch, fruit, and liquids [7,15,17,18,26]	Not all necessary texture levels were achieved for every food class, especially meat [15,17]
Hydrocolloids can be added for ink optimization and texture alteration, which is most critical in dysphagia diets [15,17,18]	Hydrocolloids may impart a non- natural flavor to foods but are necessary to achieve desired textures and printability [6,16,18]
Three-dimensional printing presents more creative freedom compared to previous methods, such as molding [7]	
Three-dimensionally printed foods can be easily fortified because they are broken down and mixed before printing [4]	
Four-dimensional food printing can further enhance sensory aspects [27,29] and holds great creative and research potential	

**Table 3 foods-11-01789-t003:** Summary of nutritional concerns and suggested alleviation methods for 3D-printed dysphagia foods.

Concern	Alleviation Method
Loss of nutrients due to leaching [7,18]	Steaming, adding cooking water to the puree
Negative influence on gut microbiome [32,34]	Adding prebiotics and probiotics
Rapid digestion of ultra-processed food [35]	Adding hydrocolloids to increase stomach viscosity
Thermal damage to the nutrients during printing [15,17]	Using low heat or room temperature
Lacking enough nutrients [4]	Fortification, using milk instead of water, adding fiber and protein

## Data Availability

Data is contained within the article.

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
