# Peer review of "3D Food Printing Applications Related to Dysphagia: A Narrative Review"

_foods, 2022, doi:10.3390/foods11121789_

Round 1
Reviewer 1 Report
In this article the authors present in a narrative manner the state of the art in 3D food printing for dysphagia applications.
The article is well written, in a clear and quite interesting way for the reader. Its content is relevant for the field and it is presented in a well-structured manner, but no original research from the the authors is included, beyond the bibliographic search of sources.
This work stablishes a good starting point for deeper and more specific research lines, and undobtedly will be relevant for an interested reader.
I just have a suggestion to the authors:
- Why choosing to present the article as a narrative review? If it was presented as a sistematic review, the paper would be much more powerful, in my humble opinion.
Congrats to the authors for their great job.
Author Response
Thank you for your kind comments and suggestions.
Regarding the systematic review suggestion, as 3D printing dysphagia foods is a new area of research and there are not enough publications with comparable study designs in order to constitute a systematic review, therefore a narrative review is more appropriate as the aim of the paper is to provide a general overview of current advances in this area. Surely in the near future as more research articles in this area are published, a systematic review would be a great idea to reach an exhaustive summary of the data.
Reviewer 2 Report
This review article provides information from 42 scientific articles about dysphagia, food categories for dysphagia patients and food printing. Also, food safety aspects of 3D printed foodstuffs are discussed. The interest for adapted food for patients with dysphagia is growing as the population is aging and the number of such people increases. Technology of 3D printing for special food is becoming more common, so the article is really useful for scientists and food technologists. The review article provides generalize, clear, easy to read and understand information, provides a perspective for future research.
Author Response
Thank you for your kind comments.
Reviewer 3 Report
The article "3D Food Printing Applications Related to Dysphagia: a Narrative Review" presents an interesting review that requires some improvements:
- Please include in the introduction section a figure that allows the reader to understand more quickly the objective and scope of the manuscript.
-Section 2.1: include a summary table that includes the advantages and disadvantages of 3D printing for dysphagia.
- Section 4: This section is very important and requires further analysis and more examples, ideally including an explanatory figure.
- Please include more background and take home for the reader, so that they can delve deeper and quickly find some relevant topics such as some elements of dysphagia as well as background on 3D printing.
Author Response
- Comment: Please include in the introduction section a figure that allows the reader to understand more quickly the objective and scope of the manuscript.
- Response: Thank you for your kind comment. Figure 2 is added as quick visual summary of how 3D printing can help creating more appealing dysphagia foods.
- Comment: Section 2.1: include a summary table that includes the advantages and disadvantages of 3D printing for dysphagia.
- Response: Thank you for your kind comment. Table 2 is added as a summary of advantages and disadvantages of 3D food printing for dysphagia.
- Comment: Section 4: This section is very important and requires further analysis and more examples, ideally including an explanatory figure.
- Response: Thank you for your kind comment. Table 3 is added as summary of nutritional concerns and suggested alleviation methods for 3D printed dysphagia foods.
- Comment: Please include more background and take home for the reader, so that they can delve deeper and quickly find some relevant topics such as some elements of dysphagia as well as background on 3D printing.
- Response: Thank you for your kind comment. More background information on dysphagia and 3D printing are added to sections 1 and 2 respectively (red font color is used to highlight the changes).
Round 2
Reviewer 3 Report
Accept in present form